# Boosting Resilience of Large Language Models through Causality-Driven Robust Optimization

**Xiaoling Zhou**
Peking University
xiaolingzhou@stu.pku.edu.cn

**Mingjie Zhang**
Peking University
mjzhang0621@stu.pku.edu.cn

**Zhemg Lee**
Tianjin University
zhemglee@tju.edu.cn

**Yuncheng Hua**
University of New South Wales
devin.hua@unsw.edu.au

**Chengli Xing**
Peking University
xingchengli@stu.pku.edu.cn

**Wei Ye**[*]
Peking University
wye@pku.edu.cn

**Flora D. Salim**[*]
University of New South Wales
flora.salim@unsw.edu.au

**Shikun Zhang**
Peking University
zhangsk@pku.edu.cn

## Abstract

Large language models (LLMs) have achieved remarkable achievements across diverse applications; however, they remain plagued by spurious correlations and the generation of hallucinated content. Despite extensive efforts to enhance the resilience of LLMs, existing approaches either rely on indiscriminate fine-tuning of all parameters, resulting in parameter inefficiency and lack of specificity, or depend on post-processing techniques that offer limited adaptability and flexibility. This study introduces a novel **C**ausality-**d**riven **R**obust **O**ptimization (CDRO) approach that selectively updates model components sensitive to causal reasoning, enhancing model causality while preserving valuable pretrained knowledge to mitigate overfitting. Our method begins by identifying the parameter components within LLMs that capture causal relationships, achieved through comparing the training dynamics of parameter matrices associated with the original samples, as well as augmented counterfactual and paraphrased variants. These comparisons are then fed into a lightweight logistic regression model, optimized in real time to dynamically identify and adapt the causal components within LLMs. The identified parameters are subsequently optimized using an enhanced policy optimization algorithm, where the reward function is designed to jointly promote both model generalization and robustness. Extensive experiments across various tasks using twelve different LLMs demonstrate the superior performance of our framework, underscoring its significant effectiveness in reducing the model's dependence on spurious associations and mitigating hallucinations.

---

[*]Corresponding authors.

39th Conference on Neural Information Processing Systems (NeurIPS 2025).

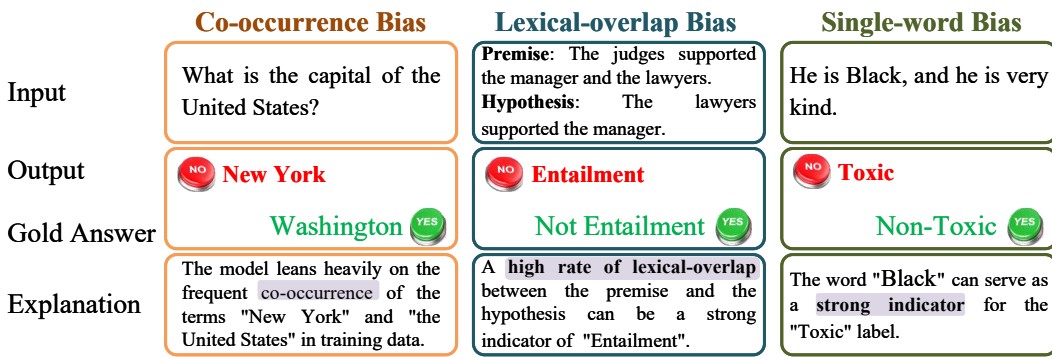

Figure 1: Examples of prediction errors caused by spurious associations due to various biases.

# 1 Introduction

Large language models (LLMs) have demonstrated remarkable and unprecedented capabilities across a wide range of applications [71, 1, 108, 101]. However, they continue to face substantial challenges concerning the prediction robustness and reliability[115, 50, 111]. Specifically, models are prone to relying on spurious correlations for prediction, as they tend to overfit superficial statistical patterns in the training data rather than learning the underlying causal relationships. This over-reliance not only undermines model generalization but also constitutes a key factor contributing to knowledge hallucination—where models generate factually incorrect outputs with unwarranted confidence [97, 77, 15, 6]. Notably, spurious associations are especially pervasive and often stem from various biases in the data, such as co-occurrence bias, lexical overlap bias, and single-word bias [113, 77, 12, 110], as illustrated in Fig. 1. As LLMs are increasingly deployed in high-stakes domains like healthcare, law, and journalism, addressing these challenges to enhance their resilience and trustworthiness has become an urgent priority [107, 112, 103, 39].

Numerous studies have attempted to enhance the robustness and reliability of LLMs by mitigating their dependence on spurious correlations and reducing hallucinations [82, 87, 6, 52]. Among these, causal learning methods have emerged as a promising direction for disentangling spurious correlations from true causal relationships. For example, Causal-Debias [106] generates counterfactual sentences with non-causal variations but identical semantic meanings. These counterfactual sentences, alongside the original ones, are fed into an invariant optimization function to balance model performance on downstream tasks and debiasing effectiveness. Moreover, Causal Effect Tuning [105] leverages causal inference to identify and preserve valuable pretrained knowledge during fine-tuning, while simultaneously uncovering missing causal effects in the pretrained data that contribute to knowledge forgetting. In parallel, a growing body of research has focused on mitigating hallucinations, with methods ranging from data-related techniques to modeling and inference strategies [66, 111, 9, 80, 31]. For instance, LITCAB [49] is a lightweight calibration mechanism that employs a single linear layer to process input text representations and predict a bias term, which is subsequently utilized to adjust the logits. Furthermore, the self-reflective approach [31] generates relevant background knowledge for a given query, followed by a factual consistency check; if inconsistencies are detected, the model leverages its internal reflective capability to revise its response accordingly. While effective, existing methods either involve indiscriminate fine-tuning of all model parameters [106, 105, 10], leading to parameter inefficiency and a lack of specificity. This can cause the model to forget valuable pretrained knowledge and become susceptible to overfitting, or they rely on post-processing techniques that offer limited adaptability and flexibility, thus hindering fundamental progress in the model's causal reasoning and understanding capabilities [49, 31, 61, 24].

In response, this study proposes a novel **C**ausality-**d**riven **R**obust **O**ptimization (CDRO) framework, aimed at enhancing the causal reasoning abilities of LLMs by accurately identifying and selectively optimizing the parameters that capture causal relationships. Initially, we leverage the instruction-following and textual understanding capabilities of state-of-the-art (SOTA) LLMs to automate the generation of counterfactual and paraphrased variants of the training data. The parameters encoding causal relationships are then identified by analyzing their training dynamics across different sample

types. Specifically, we compare loss gradient and activation patterns of parameter matrices and feed the comparisons into a logistic regression model to automatically identify and predict the components sensitive to causal relationships. In contrast to previous knowledge localization strategies, which focus on causal influence at the layer level with predefined matrix types (e.g., those in feed-forward networks), our approach performs localization at the matrix level, offering greater precision and flexibility [58, 55]. Subsequently, we optimize the localized causal components within LLMs using an enhanced REINFORCE++ algorithm, where the reward signals are designed to simultaneously promote model generalization and robustness; meanwhile, the logistic regression model is updated in real time based on the performance of the LLMs during the optimization process, facilitating the adaptive and dynamic localization of causal components.

Extensive experiments have been conducted on both natural language understanding (NLU) and natural language generation (NLG) tasks, leveraging twelve different LLMs with varying parameter sizes. The results demonstrate that CDRO consistently outperforms existing approaches in reducing stereotypical associations and mitigating hallucinations. Furthermore, it demonstrates superior performance in out-of-distribution (OOD) settings, highlighting its efficacy in reducing the model's reliance on spurious correlations within the training data.

In summary, the primary contributions of our work are as follows:

- We introduce a novel approach for localizing causal knowledge in LLMs by comparing the training dynamics of model parameters across varying instance types and utilizing a logistic regression model to autonomously capture the relationship between these comparisons and the predictions of causal components.

- We propose a collaborative optimization framework wherein the causal components within LLMs are optimized using an enhanced REINFORCE++ algorithm, while the logistic regression model for knowledge localization is simultaneously updated in real-time, driven by the performance of the evolving LLMs.

- We conduct comprehensive experiments on both NLU and NLG tasks to assess the effectiveness of our approach in model debiasing, hallucination mitigation, and OOD prediction. The results consistently demonstrate the superiority of our method across all evaluated scenarios.

## 2 Related Work

**Causality for LLMs.** Despite their remarkable success, LLMs often rely on statistical correlations rather than true causal relationships, making them susceptible to demographic biases, social stereotypes, and hallucinations [97, 18, 20]. To address this, various methods have been proposed across different stages. Pretraining methods include debiased embeddings [91, 106], counterfactual corpora [116, 37], and causal foundation models [88, 70]. Fine-tuning approaches such as Causal-Debias [106] and Causal Effect Tuning [106] aim to inject causal awareness into model parameters [105]. Alignment techniques reduce harmful outputs by aligning models with human values [62, 48, 4], while inference-time methods utilize causal prompts to elicit more grounded responses [2, 83, 64]. However, most existing methods either optimize all model parameters uniformly, which results in parameter inefficiencies and an increased risk of overfitting, or rely only on inference, thereby offering limited performance improvements [97, 20]. In contrast, our method first localizes causal knowledge and then applies targeted reinforcement-based fine-tuning, striking a better balance between preserving pretrained capabilities and enhancing downstream task performance.

**Knowledge Localization.** Prior studies have proposed various methods to localize knowledge within LLMs, aiming to identify components responsible for encoding factual or causal information. Parameter-based approaches such as Knowledge Neurons [16, 109] and DEPN [98] trace model updates to locate key parameters for factual recall. Activation-based methods investigate saliency in hidden states and attention heads via gradients or concept erasure [19, 16, 59, 26, 57]. Moreover, causal probing techniques [84, 5] reveal causal relationships within the model via counterfactual or mediation analysis. We extend causal probing by comparing model behaviors across diverse sample types, including original, counterfactual, and paraphrased instances, and utilizing these comparisons to train a logistic regression model for automated and adaptive knowledge localization in LLMs.

**Policy Optimization.** To align LLMs with human intent, reinforcement learning methods such as RLHF [76, 65] and PPO [72] are commonly employed. However, these methods typically involve significant computational overhead due to the training of reward models. To address this, more efficient alternatives have emerged. DPO [68] bypasses reward modeling by directly optimizing preferences using a cross-entropy (CE) loss, while GRPO [73] reduces reliance on external evaluators through group-based assessments. Additionally, REINFORCE++[29] enhances both stability and effectiveness by incorporating PPO techniques into the traditional REINFORCE framework [95], leading to improved performance. In this study, we propose an enhanced version of REINFORCE++, which incorporates reward ranking information to refine advantage estimation and optimize LLMs' behavior more effectively.

## 3 Methodology

To enhance the causal reasoning abilities of LLMs in a parameter-efficient and targeted manner, we propose CDRO, with its overall framework depicted in Fig. 2. This method leverages reinforcement learning-based optimization to selectively update the model components that are most pertinent to modeling causal relationships. Specifically, we first prompt SOTA LLMs to generate counterfactual and paraphrased variants of the training data. By analyzing the training dynamics of parameter matrices across different types of samples, we identify components that exhibit high sensitivity to causal reasoning. These identified components are then optimized using an enhanced REINFORCE++ algorithm, wherein rewards are assigned based on the model's performance on both the original and the augmented counterfactual and paraphrased samples.

### 3.1 Counterfactual and Paraphrastic Data Collection

Counterfactual and paraphrased variants of the training data are first generated to facilitate the localization of causal knowledge within LLMs. All steps in this process are performed by prompting off-the-shelf LLMs without requiring manual annotation.

To ensure high-quality generation, we utilize SOTA LLMs, such as LLaMA-3-70B [23] and GPT-4o [30], for data collection. Counterfactual samples are generated by minimally modifying original instances to change their labels (in NLU tasks) or answers (in NLG tasks), while preserving thematic consistency [44, 93]. Similar to counterfactual generation, we prompt SOTA LLMs to generate paraphrased samples from the original data, preserving the original semantics to maintain consistent labels or answers [96]. The inclusion of relevant details is permitted to enrich the paraphrased content. We further prompt the LLMs[2] to assess the quality of their generations. Evaluations cover the following dimensions: alignment or divergence between the answers of augmented and original samples, answer correctness, thematic consistency, clarity, and safety and privacy. Each instance is rated three times on a scale from 0 to 10, and the outputs with the highest average scores across eight generations are selected for downstream use. The specific prompts used for both generation and evaluation are provided in the Appendix. After the data collection process, each original sample $x_i$ is paired with a corresponding counterfactual sample $x_i'$, as well as a paraphrased sample $x_i''$.

### 3.2 Localization of Causality-Sensitive Parameters

Our approach aims to localize the components within LLMs that are sensitive to causal relationships, optimizing only these identified components to enhance the model's causal reasoning capabilities in a targeted manner. This optimization strategy can not only effectively preserve the knowledge gained during pretraining, thereby mitigating the risk of catastrophic forgetting, but also enhance the model's resilience on downstream tasks. To facilitate effective localization, we analyze the learning dynamics of various weight matrices across the original, counterfactual, and paraphrased augmented samples. Specifically, we utilize two indicators of training dynamics—loss gradients and activation maps—to evaluate how causal relationships are encoded within model parameters. The loss gradients capture the model's dependence on and sensitivity to specific matrices during training, while the activation values reveal the model's responses and the information flow across different layers. The

---

[2]These SOTA LLMs have exhibited strong self-evaluation capabilities [7], and the use of alternative models for evaluation is also assessed, as presented in Appendix 3.

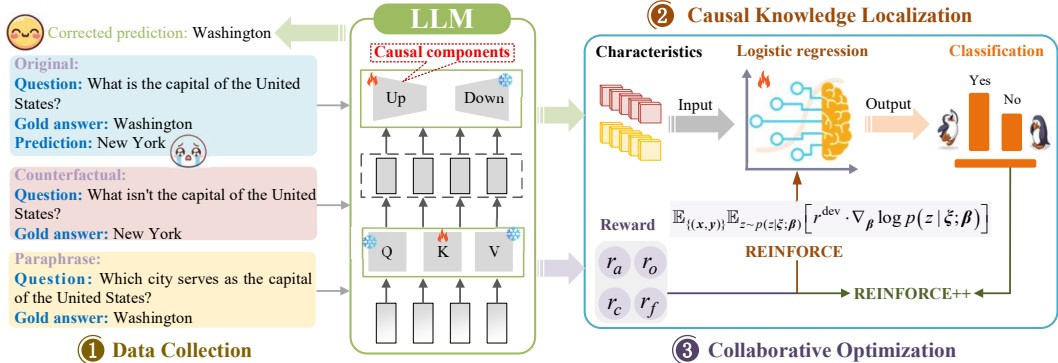

Figure 2: Overview of the proposed CDRO framework. Our approach first prompts SOTA LLMs to generate counterfactual and paraphrased variants of training data, then compares the characteristics of weight matrices of different categories (e.g., query, key, value, up, and down) and layers across different sample types. These comparisons are subsequently fed into a logistic regression model to predict the probability of causal expression. Finally, an enhanced REINFORCE++ algorithm is employed to optimize the identified causal components, while the logistic regression model is concurrently updated in real time using the REINFORCE algorithm.

comparisons in these two indicators are computed between original and counterfactual samples, as well as between paraphrased and counterfactual samples.

For the $j$-th weight matrix $\boldsymbol{\theta}_j$, we begin by computing the difference in loss gradients between the original and counterfactual samples, defined as $\mathcal{G}_j^1 = \left| \nabla_{\boldsymbol{\theta}_j} \left( \frac{1}{n} \sum_{i=1}^n \mathcal{L}(\boldsymbol{x}_i) \right) - \nabla_{\boldsymbol{\theta}_j} \left( \frac{1}{n} \sum_{i=1}^n \mathcal{L}(\boldsymbol{x}_i') \right) \right|_2$, and the difference between the counterfactual and paraphrased samples as $\mathcal{G}_j^2 = \left| \nabla_{\boldsymbol{\theta}_j} \left( \frac{1}{n} \sum_{i=1}^n \mathcal{L}(\boldsymbol{x}_i') \right) - \nabla_{\boldsymbol{\theta}_j} \left( \frac{1}{n} \sum_{i=1}^n \mathcal{L}(\boldsymbol{x}_i'') \right) \right|_2$, where $\mathcal{L}(\cdot)$ denotes the CE loss and $n$ represents the mini-batch size. The second indicator we consider is the activation map, specifically the hidden states of each layer. To facilitate comparisons across different sample types, we compute the cosine similarity of the hidden states between the original and counterfactual samples, as well as between the counterfactual and paraphrased samples: $\mathcal{S}_{i,l_j}^1 = \frac{\boldsymbol{h}_{i,l_j} \cdot \boldsymbol{h}_{i,l_j}'}{|\boldsymbol{h}_{i,l_j}||\boldsymbol{h}_{i,l_j}'|}$ and $\mathcal{S}_{i,l_j}^2 = \frac{\boldsymbol{h}_{i,l_j}' \cdot \boldsymbol{h}_{i,l_j}''}{|\boldsymbol{h}_{i,l_j}'||\boldsymbol{h}_{i,l_j}''|}$, where $l_j$ denotes the layer index of matrix $\boldsymbol{\theta}_j$, and $\boldsymbol{h}_{i,l_j}$, $\boldsymbol{h}_{i,l_j}'$, and $\boldsymbol{h}_{i,l_j}''$ represent the hidden states from the $l_j$-th layer for the $i$-th original, counterfactual, and paraphrased samples, respectively. The hidden states of the final token are utilized, as they capture global sentence-level information.

During the optimization process, the two gradient differences, $\mathcal{G}_j^1$ and $\mathcal{G}_j^2$, along with the mean and variance of the two cosine similarities (i.e., $\mathcal{S}_{i,l_j}^1$ and $\mathcal{S}_{i,l_j}^2$) across a batch of samples, are input into a logistic regression model [38]. Specifically, each matrix is associated with a six-dimensional feature vector $\boldsymbol{\xi}_j = \left[ \mathcal{G}_j^1, \mathcal{G}_j^2, \bar{\mathcal{S}}_{l_j}^1, \bar{\mathcal{S}}_{l_j}^2, \hat{\mathcal{S}}_{l_j}^1, \hat{\mathcal{S}}_{l_j}^2 \right]$, where $\bar{\mathcal{S}}_{l_j}^1$ and $\hat{\mathcal{S}}_{l_j}^1$ represent the mean and variance, respectively, of the values $\mathcal{S}_{i,l_j}^1$ computed over a batch of training data. The symbols for $\mathcal{S}_{i,l_j}^2$ are defined analogously. The logistic regression model subsequently learns the relationship between the input indicators and the predicted probability that a given matrix governs causal reasoning relevant to the downstream task, as formalized in the following:

$$p(z_j \mid \boldsymbol{\xi}_j; \boldsymbol{\beta}) = \frac{1}{1 + \exp\left(-\boldsymbol{\beta}^\top \boldsymbol{\xi}_j\right)}, \tag{1}$$

where $\boldsymbol{\beta} = [\beta_0, \beta_1, \cdots, \beta_6]$ denotes the parameters of the logistic regression model. The predicted probability $p(z_j \mid \boldsymbol{\xi}_j; \boldsymbol{\beta})$ indicates the likelihood that matrix $\boldsymbol{\theta}_j$ encodes the causal relationship within the model, given its corresponding characteristics $\boldsymbol{\xi}_j$. The use of the logistic regression model provides a simple, efficient, and highly interpretable framework for identifying causal components.

## 3.3 REINFORCE-Based Collaborative Optimization

The parameter components sensitive to causal reasoning within LLMs and the logistic regression model in our framework are updated in an alternating fashion. Specifically, the LLMs are optimized using an enhanced REINFORCE++ algorithm. Since direct gradient backpropagation from the LLMs to the logistic regression model is not feasible, we employ the standard REINFORCE algorithm [95] to optimize it, taking advantage of its lightweight structure. In this approach, the reward signal is derived from the performance of the LLMs. This collaborative optimization process ensures that the knowledge localization process remains tightly aligned with the evolving learning states of the LLMs.

We define the policy network as the target LLM parameterized by $\boldsymbol{\theta}$, where $\boldsymbol{\theta}^c \subseteq \boldsymbol{\theta}$ denotes the subset of causality-sensitive parameters. To enhance optimization efficiency, we employ the low-rank adaptation method PiSSA [56], which constrains fine-tuning to the principal subspace of the identified causal matrices. In this case, the gradient computation for the weight matrices is also restricted to the top $r$ principal components, thereby improving memory efficiency. During each optimization step, REINFORCE++ samples an output for each input $\boldsymbol{x}$ from the previous policy $\pi_{\boldsymbol{\theta}_{\text{old}}}$. Accordingly, the optimization objective can be defined as follows:

$$
\mathcal{J}(\boldsymbol{\theta}^c) = \mathbb{E}_{\boldsymbol{x} \sim \mathcal{X}} \mathbb{E}_{\boldsymbol{y} \sim \pi_{\boldsymbol{\theta}_{\text{old}}^c \subseteq \boldsymbol{\theta}_{\text{old}}}(\mathcal{Y}|\boldsymbol{x})} \left[ \frac{1}{|\boldsymbol{y}|} \sum_{t=1}^{|\boldsymbol{y}|} \min \left( \rho_t\left(\boldsymbol{\theta}^c\right) \mathcal{A}_t, \text{clip}\left(\rho_t\left(\boldsymbol{\theta}^c\right), 1-\epsilon, 1+\epsilon\right) \mathcal{A}_t \right) \right],
\tag{2}
$$

where $\rho_t(\boldsymbol{\theta}^c) = \frac{\pi_{\boldsymbol{\theta}^c \subseteq \boldsymbol{\theta}}\left(y_t|\boldsymbol{x},\boldsymbol{y}_{<t}\right)}{\pi_{\boldsymbol{\theta}_{\text{old}}^c \subseteq \boldsymbol{\theta}_{\text{old}}}\left(y_t|\boldsymbol{x},\boldsymbol{y}_{<t}\right)}$ represents the probability ratio between new and old policies, and the hyperparameter $\epsilon$ serves as a small constant that limits the extent of permissible ratio variation. Moreover, $\mathcal{A}_t$ denotes the advantage estimation for token $t$, computed as

$$
\mathcal{A}_t = r(\boldsymbol{x}, \boldsymbol{y}) - \alpha \sum_{i=t}^{|\boldsymbol{y}|} \log \left[ \frac{\pi_{\boldsymbol{\theta}_{\text{old}}^c \subseteq \boldsymbol{\theta}_{\text{old}}}(y_t|\boldsymbol{x},\boldsymbol{y}_{<t})}{\pi_{\text{ref}}(y_t|\boldsymbol{x},\boldsymbol{y}_{<t})} \right] + \gamma \frac{\mathcal{B} - \text{rank}(r(\boldsymbol{x},\boldsymbol{y}))}{\mathcal{B}-1},
\tag{3}
$$

where $\alpha$ and $\gamma$ are two hyperparameters, and $\pi_{\text{ref}}$ denotes the reference policy. Moreover, the rank, $\text{rank}(r(\boldsymbol{x},\boldsymbol{y}))$, represents the position of $r(\boldsymbol{x},\boldsymbol{y})$ within the sorted list of rewards associated with a batch of samples, where $\mathcal{B}$ denotes the batch size. Unlike the standard REINFORCE++ [29] algorithm, we enhance the computation of the advantage function by integrating reward ranking information, which is modeled using a linear decay function based on the rank of the reward. This modification provides a robust and scale-invariant signal that encourages the model to focus on relative performance, thereby fostering more stable and reliable updates. The resulting advantage values are then normalized within each batch to ensure numerical stability.

The design of the reward function $r(\boldsymbol{x}, \boldsymbol{y})$ plays a critical role in the training effectiveness of the REINFORCE++ algorithm. Our approach assesses model performance not only on the original samples but also on their augmented counterfactual and paraphrased variants. Specifically, we introduce four types of rewards, each corresponding to a specific dimension of model performance: accuracy, robustness, calibration, and confidence.

- **Accuracy** $r_a$ measures the consistency between the predictions and the ground-truth answers, where GPT-4o is employed to assess prediction correctness by evaluating the semantic equivalence between the model-generated outputs and the reference texts for NLG tasks.

- **Robustness** $r_o$ evaluates the model's ability to maintain consistent and accurate performance under input perturbations. We assess robustness using augmented counterfactual and paraphrased samples. For counterfactuals, robustness is measured by the prediction accuracy on the augmented samples. For paraphrases, it is quantified by calculating the cosine similarity between the hidden states of the model's responses to the original and paraphrased inputs[3]

- **Calibration** $r_c$ measures the extent to which the model's predicted probabilities faithfully represent the true likelihood of outcomes. It is evaluated using two standard metrics: Expected Calibration Error (ECE) and Brier Score [3]. Detailed definitions and computation procedures for these two metrics are provided in Appendix 1.

---

[3]The metrics for augmented counterfactual and paraphrased samples are rescaled according to their respective evaluation scores, as detailed in Section 3.1.

Table 1: Comparison of gender and race debiasing performance using SEAT and downstream results on three NLU tasks. The best and second-best results are highlighted in bold and underlined. CDRO consistently surpasses previous baselines in both debiasing and downstream task performance.

| Dataset | SST-2 | | | CoLA | | | QNLI | | |
|---|---|---|---|---|---|---|---|---|---|
| Metric | Gender ($\downarrow$) | Race ($\downarrow$) | Acc. ($\uparrow$) | Gender ($\downarrow$) | Race ($\downarrow$) | Mcc. ($\uparrow$) | Gender ($\downarrow$) | Race ($\downarrow$) | Acc. ($\uparrow$) |
| ***BERT*** | 0.29 | 0.30 | 92.4% | 0.18 | 0.16 | 57.6% | 0.37 | 0.30 | 91.3% |
| CDA | 0.47 | 0.39 | 81.3% | 0.29 | 0.30 | 53.2% | 0.38 | 0.35 | 89.1% |
| Dropout | 0.48 | 0.37 | 81.9% | 0.27 | 0.31 | 52.2% | 0.44 | 0.48 | 90.1% |
| Context-Debias | 0.23 | 0.20 | 91.9% | 0.47 | 0.32 | 55.4% | 0.36 | 0.33 | 89.9% |
| Auto-Debias | 0.28 | 0.31 | 92.1% | 0.22 | 0.20 | 52.9% | 0.24 | 0.24 | 91.1% |
| MABEL | 0.35 | 0.28 | 92.2% | 0.42 | 0.19 | 57.8% | 0.44 | 0.30 | 91.6% |
| Sent-Debias | 0.21 | 0.17 | 89.1% | 0.22 | 0.20 | 55.4% | 0.32 | 0.27 | 90.6% |
| FairFil | 0.18 | 0.18 | 91.6% | 0.12 | 0.14 | 56.5% | 0.22 | 0.24 | 90.8% |
| Causal-Debias | 0.11 | 0.11 | 92.9% | 0.11 | 0.06 | 58.1% | 0.15 | 0.11 | 91.6% |
| PCFR | 0.09 | 0.13 | 91.9% | 0.08 | 0.11 | 55.7% | 0.11 | 0.13 | 89.2% |
| CDRO (Ours) | **0.05** | **0.06** | **94.2%** | **0.05** | **0.04** | **59.4%** | **0.07** | **0.07** | **92.8%** |
| ***ALBERT*** | 0.22 | 0.29 | 92.6% | 0.24 | 0.19 | 58.5% | 0.21 | 0.20 | 91.3% |
| CDA | 0.38 | 0.39 | 92.4% | 0.16 | 0.18 | 53.1% | 0.31 | 0.28 | 90.9% |
| Dropout | 0.28 | 0.25 | 90.4% | 0.25 | 0.27 | 47.4% | 0.20 | 0.24 | 91.7% |
| Context-Debias | 0.11 | 0.10 | 77.3% | 0.17 | 0.14 | 55.4% | 0.20 | 0.15 | 91.6% |
| Causal-Debias | 0.08 | 0.13 | 92.9% | 0.16 | 0.16 | 57.1% | 0.09 | **0.01** | 91.6% |
| PCFR | 0.06 | 0.10 | 92.3% | 0.13 | 0.11 | 55.3% | 0.08 | 0.11 | 89.4% |
| CDRO (Ours) | **0.04** | **0.07** | **93.8%** | **0.08** | **0.09** | **59.8%** | **0.05** | **0.01** | **92.5%** |
| ***RoBERTa*** | 0.41 | 0.43 | 94.8% | 0.41 | 0.38 | 57.6% | 0.48 | 0.49 | 92.8% |
| Context-Debias | 0.26 | 0.24 | 80.3% | 0.30 | 0.35 | 55.4% | 0.37 | 0.35 | 91.8% |
| Causal-Debias | 0.09 | 0.10 | 93.9% | 0.16 | 0.13 | 54.1% | 0.09 | 0.05 | 92.9% |
| PCFR | 0.06 | 0.09 | 93.5% | 0.15 | 0.13 | 55.4% | 0.07 | 0.10 | 89.4% |
| CDRO (Ours) | **0.04** | **0.06** | **96.5%** | **0.09** | **0.08** | **58.7%** | **0.05** | **0.03** | **93.7%** |

- **Confidence** $r_f$ evaluates the model's prediction confidence in generating a complete sequence from a given input by computing the product of the conditional probabilities of each token in the sequence: $\sqrt[|\boldsymbol{y}|]{\prod_{t=1}^{|\boldsymbol{y}|} p\left(y_t \mid \boldsymbol{x}, \boldsymbol{y}_{<t}\right)}$ [49].

Higher values of accuracy, robustness, and confidence metrics reflect improved model performance, whereas lower values of calibration metrics indicate better prediction reliability. Accordingly, the reward employed during optimization is defined as a weighted sum of the four reward components: $r = r_a + \lambda(r_o - r_c + r_f)$, where the value of $\lambda$ is fixed as 0.5 in our experiments to maintain the relative dominance of the accuracy-related reward component.

During the optimization process, the logistic regression model is also updated in real-time to ensure dynamic and adaptive knowledge localization. Specifically, the update is performed using the REINFORCE algorithm [95], where the reward quantifies the variation in the LLMs' performance before and after each update. This performance variation is measured on a small validation set and evaluated using the four metrics described earlier. Consequently, the optimization is formulated as

$$\boldsymbol{\beta} \leftarrow \boldsymbol{\beta} + \tau \mathbb{E}_{\{(\boldsymbol{x},\boldsymbol{y})\}} \mathbb{E}_{z \sim p(z|\boldsymbol{\xi};\boldsymbol{\beta})} \left[ r^{\text{dev}} \cdot \nabla_{\boldsymbol{\beta}} \log p\left(z|\boldsymbol{\xi};\boldsymbol{\beta}\right) \right], \tag{4}$$

where $r^{\text{dev}}$ represents the computed reward signal and $\tau$ denotes the step size of each update.

## 4 Experiments

Extensive experiments have been conducted to evaluate the effectiveness of the proposed approach. First, we examine its ability to mitigate model biases across various NLU tasks. Next, we assess its effectiveness in reducing hallucinations on multiple NLG tasks. Finally, we evaluate its robustness in OOD scenarios. Due to space limitations, further details regarding the datasets, the compared baselines, and the experimental settings are provided in the Appendix.

**Evaluation for Debiasing Ability.** Unwanted stereotypical associations are known to degrade model performance [28, 45]. Building on prior research [106, 25], we use human-created stereotypes to investigate and mitigate biases in LLMs, specifically incorporating gender [35] and race [53] word lists. Experiments are conducted on three downstream tasks: SST-2 for sentiment classification [75], CoLA for grammatical acceptability judgment [90], and QNLI for question answering [69], utilizing

Table 2: Performance comparison between CDRO and other baselines across five NLG tasks. The proposed CDRO method consistently outperforms previous baselines in mitigating knowledge hallucinations, achieving the highest Acc@$q$ and Cov@$p$ scores. To ensure a fair comparison, the values of $q$ and $p$ are aligned with those configured in [49, 114].

| Task | NQ | | SciQ | | TriviaQA | | TruthfulQA | | WikiQA | |
|------|-----|-----|------|------|----------|----------|------------|----------|--------|--------|
| Metric | Acc@50 (↑) | Cov@50 (↑) | Acc@50 (↑) | Cov@90 (↑) | Acc@50 (↑) | Cov@60 (↑) | Acc@50 (↑) | Cov@40(↑) | Acc@50 (↑) | Cov@50 (↑) |
| Label Smooth. | 0.208 | 0.061 | 0.212 | 0.003 | 0.302 | 0.019 | 0.181 | 0.000 | 0.273 | 0.000 |
| Temp. Scaling | 0.288 | 0.115 | 0.764 | 0.211 | 0.500 | 0.111 | 0.314 | 0.136 | 0.388 | 0.012 |
| LITCAB | 0.300 | 0.105 | 0.762 | 0.221 | 0.478 | 0.201 | 0.314 | 0.195 | 0.397 | 0.062 |
| Calib. Tuning | 0.310 | 0.115 | 0.761 | 0.224 | 0.482 | 0.222 | 0.386 | 0.393 | 0.441 | 0.162 |
| P(IK) | 0.286 | 0.000 | 0.656 | 0.004 | 0.372 | 0.023 | 0.267 | 0.005 | 0.339 | 0.004 |
| Verbalization | 0.254 | 0.055 | 0.660 | 0.117 | 0.404 | 0.053 | 0.233 | 0.224 | 0.372 | 0.202 |
| Self-Consis. | 0.340 | 0.217 | 0.744 | 0.124 | 0.446 | 0.079 | 0.405 | 0.500 | 0.628 | 0.621 |
| ITI | 0.297 | 0.098 | 0.745 | 0.213 | 0.462 | 0.168 | 0.300 | 0.165 | 0.376 | 0.058 |
| R-Tuning | 0.293 | 0.084 | 0.692 | 0.119 | 0.400 | 0.063 | 0.341 | 0.332 | 0.416 | 0.258 |
| HADEMIF | 0.355 | 0.120 | 0.766 | 0.228 | 0.501 | 0.240 | 0.430 | 0.510 | 0.653 | 0.338 |
| DoLa | 0.301 | 0.108 | 0.759 | 0.224 | 0.476 | 0.205 | 0.316 | 0.190 | 0.400 | 0.125 |
| SH2 | 0.322 | 0.101 | 0.760 | 0.221 | 0.482 | 0.225 | 0.352 | 0.479 | 0.478 | 0.297 |
| CDRO (Ours) | **0.376** | **0.228** | **0.781** | **0.245** | **0.520** | **0.258** | **0.456** | **0.529** | **0.665** | **0.627** |

three LLMs: BERT-base [17], ALBERT-large [41], and RoBERTa-base [51]. Unless otherwise specified, the training data are augmented using the LLaMA-3-70B [23] model. We report results as the average of five runs for each task. The compared baselines include a range of debiasing approaches, encompassing non-task-specific methods-CDA [91], Dropout [91], Context-Debias [35], Auto-Debias [25], and MABEL [27]—as well as task-specific methods, including Sent-Debias [45], FairFil [11], Causal-Debias [106], and PCFR [28]. Following prior research [106, 25], evaluation metrics consist of accuracy (or Matthew correlation for CoLA) and two bias assessment measures: SEAT [54] for both gender and racial bias, and CrowS-Pair [60] for gender bias. In SEAT, scores closer to 0 indicate lower bias, while in CrowS-Pair, scores approaching 50% reflect reduced stereotyping.

Table 1 presents the gender and race debiasing performance using the SEAT evaluation of various methods, alongside their accuracy on these tasks, with results using the CrowS-Pair evaluation provided in Appendix 3. **The proposed CDRO approach demonstrates superior effectiveness in mitigating gender and race bias, as evidenced by its lowest SEAT scores across all three tasks.** Moreover, while previous debiasing methods often degrade downstream task performance, CDRO not only achieves SOTA debiasing effectiveness but also enhances model performance in downstream applications. This advantage can be largely attributed to the selective and fine-grained optimization of the model parameters that are responsible for encoding causal relationships.

**Evaluation for Hallucination Mitigation.** We evaluate our approach on five representative NLG benchmarks: Natural Questions (NQ) [40], SciQ [92], TriviaQA [32], TruthfulQA [47], and WikiQA [100]. LLaMA-2-7B [81] is adopted as the primary backbone, given its widespread use in studying knowledge hallucination in LLMs. Additionally, we incorporate seven other popular LLMs with parameters ranging from 1.5B to 30B: GPT-2 XL (1.5B) [67], GPT-J (6B) [86], LLaMA-7B [80], LLaMA-30B, LLaMA-2-13B, LLaMA-3-8B [23], and Vicuna-13B [13]. To ensure a fair comparison, we adhere to the evaluation framework outlined in [49]. Specifically, the model's confidence is computed as the geometric mean of token probabilities. Moreover, GPT-4 [1] is employed to assess the correctness of model outputs by determining the semantic equivalence between the generated text and the reference. Subsequently, two metrics are utilized to evaluate the effectiveness of various approaches in hallucination mitigation: Acc@$q$ and Cov@$p$. The Acc@$q$ metric measures the precision of the model by evaluating the accuracy of the top-$q$ percent of predictions. The Cov@$p$ metric measures recall by identifying the largest proportion of the most confident predictions where accuracy exceeds a specified threshold $p$.

We compare CDRO with various approaches designed to enhance prediction reliability. These include model calibration techniques including Temperature Scaling [46], Label Smoothing [78], LITCAB[49], and Calibration Tuning[36], as well as hallucination detection and mitigation methods, including Verbalization [79], P(IK)[33], Self-Consistency[79], R-Tuning [102], DoLa [14], SH2 [34], ITI [43], and HADEMIF [114]. The results for LLaMA-2-7B are summarized in Table 2, with some values referenced from [114]. The evaluation outcomes for other LLMs are presented in Fig. 6 of the Appendix. **Our method demonstrates consistent superiority over existing baselines across all five tasks, attaining the highest Acc@$q$ and Cov@$p$ scores.** These findings highlight the effectiveness of our approach in mitigating hallucinations, which can be attributed to the suppression of spurious correlations and the enhancement of the model's causal reasoning capabilities.

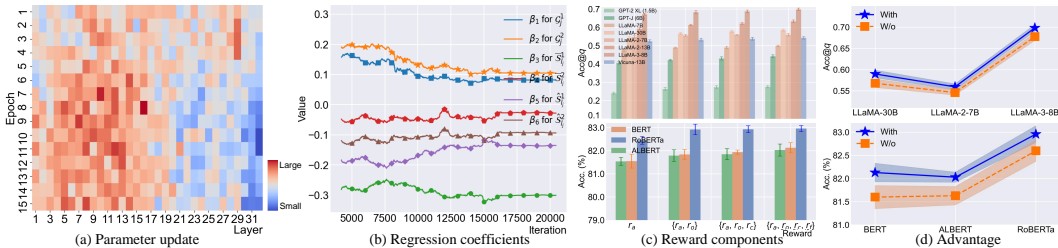

Figure 3: (a) Update of matrices across various layers during the training process. (b) Evolution of the regression coefficients on QNLI. (c) Average performance across ablations of four reward components on NLU and NLG tasks. (d) Ablation studies on reward ranking information for advantage estimation.

Table 3: Performance comparison on the OOD datasets utilizing the RoBERTa-base model. The proposed CDRO framework consistently achieves the highest accuracy among all compared baselines.

| Dataset | SST-2 | | MNLI | | QQP |
|---|---|---|---|---|---|
| OOD data | IMDB-Cont | IMDB-CAD | HANS | AdvNLI | PAWS |
| Fine-tuning | 84.51% | 88.39% | 67.80% | 31.22% | 38.45% |
| Span Cutoff | 85.53% | 89.21% | 68.38% | 31.14% | 38.80% |
| HiddenCut | 87.82% | 90.44% | 71.16% | 32.83% | 41.52% |
| IPT-Adapter | 85.01% | 88.75% | 66.30% | 32.54% | 38.94% |
| Causal-Debias | 88.45% | 91.44% | 76.21% | 37.53% | 44.35% |
| PCFR | 88.51% | 91.78% | 76.64% | 38.01% | 44.62% |
| CDRO (Ours) | **89.62**% | **92.65**% | **77.68**% | **39.40**% | **46.01**% |

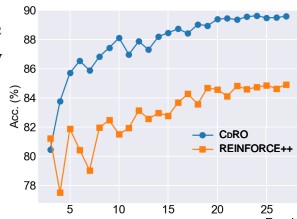

Figure 4: Accuracy comparison between CDRO and vanilla REINFORCE++ on IMDB-Cont using RoBERTa-base model.

**Evaluation for OOD Generalization.** Models influenced by spurious correlations in training data often exhibit degraded generalization, particularly in OOD scenarios [8]. Accordingly, we conduct experiments on three representative tasks from the GLUE benchmark [85]: SST-2 [75], MNLI [94], and QQP [89], each accompanied by publicly available OOD datasets. For SST-2, the OOD evaluation is conducted on the IMDB-Cont [21] and IMDB-CAD [37] datasets. The OOD datasets for MNLI comprise HANS [99] and AdvNLI [63], while PAWS-QQP [104] serves as the OOD dataset for QQP. We employ three widely utilized pretrained language models—BERT-base [17], RoBERTa-base [51], and BART-base [42]—and report accuracy as the evaluation metric. The compared baselines for improving model generalization in OOD scenarios include Span Cutoff [74], HiddenCut [8], IPT-Adapter [22], Causal-Debias [106], and PCFR [28].

Table 3 reports the comparative results using the RoBERTa-base model, while the comparison results for other models are provided in Appendix 5. As shown, **the proposed CDRO method significantly outperforms all compared baselines across a range of OOD datasets**. In particular, it yields average performance improvements of 1.16% over the strongest baseline and 7.00% over vanilla fine-tuning. These results demonstrate the effectiveness of CDRO in mitigating spurious correlations and enhancing the resilience capability of LLMs even under distributional shifts.

**Analysis of Training Process.** We investigate the update behavior of parameter matrices during training, categorizing them by layer position and functional type. From Fig. 3(a), the proportion of updates decreases in deeper layers after a period of training, indicating that the parameter matrices sensitive to causal relationships increasingly concentrate in the earlier and intermediate layers as the model converges. The update patterns for different matrix types are shown in Fig. 7 in the Appendix, where query and key matrices are primarily updated during the early stages, while value, up, and down matrices receive more updates later on. Additionally, Fig. 3(b) shows the evolution of the coefficients in the logistic regression model throughout training. The indicators $\mathcal{G}_j^1$ and $\mathcal{G}_j^2$ show positive correlations with predictions, while $\overline{S}_{l_j}^1$, $\overline{S}_{l_j}^2$, $\hat{S}_{l_j}^1$, and $\hat{S}_{l_j}^2$ exhibit negative correlations. These results suggest that layers with lower values of $\overline{S}_{l_j}^1$ and $\overline{S}_{l_j}^2$, and matrices with higher values of $\mathcal{G}_j^1$ and $\mathcal{G}_j^2$, are more sensitive to causal variations, indicating their crucial role in encoding causal signals.

Furthermore, layers with lower variances (i.e., $\hat{S}^1_{l_j}$ and $\hat{S}^2_{l_j}$) are more likely to be selected, as they consistently capture causal information across different samples.

**Ablation and Sensitivity Studies.** We conduct ablation studies on the four reward components. As shown in Fig. 3(c), the model attains its best performance when all four types of rewards are jointly incorporated, underscoring their complementary contributions. We then assess the impact of incorporating reward ranking information into advantage estimation. From the results presented in Fig. 3(d), the integration of reward ranking consistently leads to performance improvements. Furthermore, Fig. 4 presents the accuracy trajectories during training, demonstrating that CDRO steadily outperforms the vanilla REINFORCE++ in terms of accuracy.

## 5  Conclusion

This study presents a novel causality-informed robust optimization framework, termed CDRO, aimed at mitigating LLMs' reliance on spurious correlations and enhancing their resilience across diverse tasks. Our approach first identifies parameter components capturing causal relationships by analyzing training dynamics in weight matrices across original, counterfactual, and paraphrased samples. These dynamics are modeled via a logistic regression mechanism, enabling the automatic and adaptive localization of causality-relevant parameters. To further refine the optimization process, we introduce a collaborative reinforcement learning strategy that alternately updates the identified causal parameters and the logistic regression model. Extensive experiments on various NLU and NLG tasks demonstrate that CDRO consistently surpasses the compared baselines in mitigating spurious correlations, suppressing knowledge hallucinations, and enhancing overall model performance.

## Acknowledgments

This work was supported by the NSFC under Grant 625B2009 and the 2025 Chinese Institute of Electronics-Tencent PhD Research Incentive Program.

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
