# OpenReview forum: "Boosting Resilience of Large Language Models through Causality-Driven Robust Optimization"
_NeurIPS.cc/2025/Conference — NeurIPS 2025 poster_

### Official Review · Reviewer_6STb · 2025-06-26

**Clarity:** 3
**Significance:** 3
**Originality:** 3
**Rating:** 4
**Confidence:** 3

**Summary:**

The manuscript presents a novel Causality-driven Robust Optimization (CDRO) approach that selectively updates model components sensitive to causal reasoning. Extensive experiments across various tasks and LLMs architectures demonstrate the superior performance of the proposed method.

**Questions:**

1. How was the threshold determined for selecting causal components? Specifically, the appendix mentions selecting matrices with "predicted probabilities exceeding the average predicted probability," but is this optimal? Were other selection strategies considered?
2. How do you verify that the identified components truly encode causal relationships rather than other statistical patterns? Could you provide ablation studies or visualization evidence showing that these components specifically capture causal rather than correlational information?
3. How do the training time and computational requirements compare between CDRO and other methods? The dual optimization process (updating both the logistic regression model and the LLM parameters) may introduce computational complexity.

**Ethical Concerns:**

["NO or VERY MINOR ethics concerns only"]

**Final Justification:**

The authors have resolved most of my concerns.  I decide to give it a Borderline accept score.

**Limitations:**

yes

**Quality:**

3

**Strengths And Weaknesses:**

Strengths:
1. CdRO addresses a critical challenge in LLM development: enhancing causal reasoning without extensive retraining or parameter inefficiency.
2. There is a clear problem formulation that effectively motivates the need for parameter-efficient causal reasoning enhancement.
3. The manuscript has a well-structured presentation with logical flow from theoretical foundations to implementation details and experimental results.

Weaknesses:
1. The explanation of how loss gradients capture model sensitivity needs more intuitive examples or visualizations.
2. The computational overhead introduced by the knowledge localization process could present scalability challenges for very large models, though this is partially mitigated by the parameter efficiency.
3. There is limited discussion of potential negative effects or failure cases, in which the approach might underperform relative to baselines.
4. How stable is the identification of causal components across different runs or different datasets? Is there consistency in which components are identified as causality-sensitive?

---

> ### Author Rebuttal · Authors · 2025-07-31
>
> Dear Reviewer 6STb,
>
> We sincerely appreciate your insightful review and valuable comments and suggestions. Thank you very much for recognizing the novelty and significance of our research, the clarity of the problem formulation, the structured presentation, the comprehensiveness of the experiments, and the strong performance of our method. Below, we provide detailed responses to each of your concerns:
>
> > **Q1: Relationship between loss gradients and causal sensitivity.**
>
> We humbly respond to your concerns from the following three points:
> - The motivation for identifying causal-sensitive parameters stems from the idea that **if a parameter is causal-sensitive, its behavior—such as loss gradients and activation values—will differ between the original sample and its corresponding counterfactual**. This concept is akin to previous methods of parameter or activation-based localization, where model parameters or sample activations are perturbed, and changes in model predictions are analyzed for localization purposes [1,2]. However, manually determining the impact of different behavioral indicators on localization results is unreasonable. To address this, we propose utilizing a simple logistic model to automatically learn this relationship.
> - As discussed in Section 3.2, we select loss gradients and activation values because they **capture complementary aspects of model behavior**: loss gradients reflect the model’s sensitivity to specific parameter matrices during training, while activation values indicate the information flow and layer-wise responses to input data.
> - As illustrated in **Figs. 3(b) and 11**, the relationship between causal localization results and the utilized indicators aligns with our expectations. The table below further provides the converged coefficients for each indicator on three datasets. Specifically, the coefficients ($\beta_{1}$ and $\beta_{2}$) for both gradient difference indicators are positive, whereas those ($\beta_{3}$ and $\beta_{4}$) for the cosine similarity of hidden states are negative. This suggests that **larger gradient differences and lower cosine similarity of activation values are associated with a higher likelihood of a matrix being causal-sensitive**. Additionally, the coefficients ($\beta_{5}$ and $\beta_{6}$) for the variance of cosine similarities are negative, primarily because lower variance indicates more consistent capture of causal information across different samples.
> |Coefficient|$\beta_{1}$|$\beta_{2}$|$\beta_{3}$|$\beta_{4}$|$\beta_{5}$|$\beta_{6}$|
> |:-|:-:|:-:|:-:|:-:|:-:|:-:|
> |QNLI|0.080|0.103|-0.303|-0.031|-0.137|-0.097|
> |SST-2|0.117|0.110|-0.273|-0.252|-0.140|-0.165|
> |SciQ|0.070|0.105|-0.290|-0.273|-0.150|-0.130|
>
> > **Q2: Computational overhead.**
>
> As you mentioned, despite the increased time complexity introduced by the dual optimization process, our approach demonstrates **superior parameter efficiency**, thereby achieving **a better tradeoff between preserving pretrained knowledge and enhancing downstream task performance**. Moreover, it consistently **achieves the best performance across various tasks and models**. Additionally, the logistic model **converges after a limited number of training iterations** (as shown in Figs. 3(b) and 11), allowing the omission of the logistic update step but only optimizing LLMs in subsequent training stages, further enhancing training efficiency.
>
> In our framework, the LLM is optimized using the REINFORCE++ algorithm with a PiSSA-based low-rank adaptation strategy. Thus, we compare its time complexity with that of REINFORCE++ using PiSSA adaptation, which allows us to isolate the additional time introduced by our components. The results below show that **the ratio of increased time complexity relative to the baseline method decreases with larger models**. Moreover, it is worth noting that while our method incurs some additional computational overhead, it remains a highly efficient fine-tuning approach, particularly when compared to the full fine-tuning for LLMs.
>
> |Method|Time|Tunable # param.|
> |:-|:-:|:-:|
> |RoBERTa-base|||
> |REINFORCE++|591s|1.0M|
> |CDRO|1514s↑156%|0.7M↓30%|
> |LLaMA-2-7B|||
> |REINFORCE++|8750s|56.1M|
> |CDRO|11387s↑30%|23.3M↓58%|
> |LLaMA-3-8B|||
> |REINFORCE++|8924s|57.0M|
> |CDRO|11406s↑28%|23.3M↓59%|
> |Qwen2.5-32B|||
> |REINFORCE++|25700|266M|
> |CDRO|28787↑12%|93M↓65%|
>
> > **Q3: Potential negative effects or failure cases.**
>
> Although our method yields significant performance improvements across diverse scenarios, its effectiveness may degrade under inappropriate settings.
> - The quality and reliability of the generated counterfactual and paraphrased samples can affect the effectiveness of parameter localization and optimization. Using low-performing LLMs for data generation may produce low-quality augmentations that degrade localization accuracy and impair overall model performance.
> - Overly conservative thresholding in selecting causality-sensitive parameters may overly limit updates, hindering the model's adaptability to downstream tasks. To mitigate this, we adopt a dynamic thresholding strategy based on the average logistic probabilities across different matrices, which has demonstrated effectiveness across various datasets and models.
>
> The above analysis will be incorporated into our manuscript to extend the discussion of failure cases. In addition, other limitations, including inapplicability to black-box models and increased complexity, are discussed in **Appendix A.13**.
>
> > **Q4: Stability of the identified causal components.**
>
> Insightful question. The causal components identified by our method demonstrate **stability across different runs but exhibit variability across different datasets**. Nevertheless, through extensive analysis, we have uncovered two fundamental, consistent patterns, detailed in the **“Analysis of Training Process” subsection in Section 4 (Figs. 3(a) and 13)**:
> - As the model converges, parameter matrices sensitive to causal relationships increasingly concentrate in the earlier and intermediate layers.
> - During early training stages, the query and key matrices are primarily updated, whereas the value, up, and down matrices receive more updates in later stages.
>
> Moreover, we observe that **the influence of various indicators on the logistic model’s decision-making is relatively consistent across datasets (Figs. 3(b) and 11)**. Specifically, the coefficients for the two gradient difference indicators are consistently positive, while those for the cosine similarities of hidden states are persistently negative. This indicates that larger gradient differences and lower cosine similarity are associated with a higher likelihood of a matrix being causal-sensitive.
>
> > **Q5: Threshold for selecting causal components.**
>
> In our experiments, the threshold for selecting causal components is defined as the average logistic probability across all parameter matrices. This dynamic threshold enables updating matrices with probabilities exceeding the average. **Given that the localization network adapts dynamically during model training, we argue that this adaptive threshold, rather than a fixed, manually set one, can yield superior performance**. The table below compares several threshold strategies—Top-15 matrices, matrices with probabilities exceeding 0.5, and the average threshold—with the latter consistently outperforming fixed manual thresholds. Nevertheless, in practical applications, the threshold can be adjusted flexibly to meet specific requirements.
>
> |Dataset|SST-2||MNLI||QQP||
> |:-|:-:|:-:|:-:|:-:|:-:|:-:|
> |OOD data|IMDB-Cont|IMDB-CAD|HANS|AdvNLI|PAWS|
> |Average|**89.62**|**92.65**|**77.68**|**39.40**|**46.01**|
> |Top-15|89.19|92.37|77.54|38.98|45.43|
> |0.5|89.23|92.48|77.35|38.92|45.66|
>
> Additionally, **the consistent superior performance of our approach suggests that this threshold remains robust across different models and tasks**.
>
> > **Q6: Encoding causal relationships rather than statistical patterns.**
>
> Good question. We respectfully respond to your concerns from the following two points:
> - As stated in the response to Q1, the motivation for identifying causal-sensitive parameters stems from the idea that if a parameter is causal-sensitive, its behavior will differ between the original sample and its counterfactual. From the optimized coefficients shown in **Figs. 3(b) and 11**, the learned relationships between causal localization and the applied indicators align with our expectations. Specifically, the coefficients for both gradient difference indicators are positive, while those for the cosine similarity of hidden states are negative. These findings confirm that **the identified parameters behave distinctly between original and counterfactual samples, supporting their sensitivity to causal relationships according to the intuitive expectations from causal inference**.
> - Intuitively, relying solely on paraphrased and original samples may lead localized parameters to capture superficial statistical patterns rather than true causal relationships. Accordingly, we conduct ablation experiments considering either paraphrased or counterfactual samples exclusively. The results below indicate that **omitting counterfactual samples causes a significantly greater performance drop than omitting paraphrased samples, evidencing that localized parameters effectively capture causal relationships via counterfactual augmentations**.
> |Task|NQ||SciQ||TriviaQA||TruthfulQA||WikiQA||
> |:-|:-:|:-:|:-:|:-:|:-:|:-:|:-:|:-:|:-:|:-:|
> |Metric|Acc@50(↑)|Cov@50(↑)|Acc@50(↑)|Cov@90(↑)|Acc@50(↑)|Cov@60(↑)|Acc@50 (↑)|Cov@40(↑)|Acc@50 (↑)|Cov@50 (↑)|
> |CDRO|**0.376**|**0.228**|**0.781**|**0.245**|**0.520**|**0.258**|**0.456**|**0.529**|**0.665**|**0.627**|
> |W/o paraphrase|0.375|0.224|0.777|0.243|0.519|0.255|0.450|0.528|0.662|0.624|
> |W/o counterfactual|0.359|0.220|0.769|0.233|0.505|0.247|0.436|0.514|0.657|0.621|
>
> [1] Locating and editing factual associations in gpt.
>
> [2] Mass-editing memory in a transformer.

---

> ### Comment · Reviewer_6STb · 2025-08-04
>
> Thanks for your efforts; it has resolved most of my confusion. However, considering my reservations about learning and identifying more complex causal relationships using a simple logistic model in real-world scenarios, I might only give it a Borderline accept score.

---

> > ### Author Response · Authors · 2025-08-04
> > **Thank you for your timely reply.**
> >
> > Dear Reviewer 6STb,
> >
> > Thank you very much for your timely reply! We sincerely appreciate that you found our replies have addressed your concerns and that you advocate for the acceptance of our paper. We are also truly grateful for all your valuable comments and constructive suggestions, which have significantly enhanced the quality and clarity of our manuscript.
> >
> > The purpose of employing a logistic regression model in our framework is to **model the relationship between the behavioral divergence of weight matrices across original and counterfactual samples and the identification of causally sensitive parameters**, rather than to directly learn and identify causal relationships. We fully acknowledge that causal relationships in real-world scenarios are often complex. Accordingly, given that counterfactual is a crucial tool for modeling and understanding causality, our framework constructs counterfactual samples to expose behavioral differences in parameter matrices, enabling the localization of causally sensitive components, which are then selectively optimized via an enhanced policy optimization strategy. Both causal inference intuition and empirical evidence suggest that **greater behavioral divergence between original and counterfactual samples corresponds to a higher likelihood of causal sensitivity in the parameter matrix**. Consequently, modeling this relationship via logistic regression provides a simple yet principled probabilistic framework.
> >
> > Furthermore, we validate the effectiveness of our method across diverse scenarios—debiasing, hallucination mitigation, and out-of-distribution (OOD) generalization—all of which demand stronger causal reasoning and robustness. While debiasing targets specific attribute biases (e.g., gender or race), hallucination mitigation (including our original experiments and new results on two hallucination datasets: FACTOR and HaluEval-Sum) and OOD generalization involve more complex and varied causal structures. **Despite this complexity, our proposed causality-driven robust optimization framework consistently improves model performance in these challenging settings**.
> >
> > |Data|FACTOR||HaluEval||
> > |:-|:-:|:-:|:-:|:-:|
> > ||Wiki|News|Acc-A|Acc-H|
> > |LLaMA-2-7B|58.64|72.25|48.01|19.92|
> > |+DoLa|60.13|72.86|48.65|27.78|
> > |+SH2|64.10|73.59|50.49|50.50|
> > |+CDRO|**65.29**|**74.76**|**51.58**|**51.66**|
> > |LLaMA-3-8B|65.59|78.42|55.07|28.95|
> > |+DoLa|68.90|79.71|54.92|30.43|
> > |+SH2|70.03|80.55|60.38|53.81|
> > |+CDRO|**71.05**|**82.45**|**61.27**|**55.03**|
> >
> > In summary, by adaptively identifying causally sensitive parameters and incorporating robust optimization, our framework presents a practical and effective approach to substantially enhancing the causal reasoning capabilities and robustness of LLMs, while concurrently mitigating the risk of overfitting.
> >
> >
> >
> > Once again, we are sincerely grateful for your insightful comments and the pivotal role you have played in enhancing the quality of our manuscript! We deeply appreciate your recommendation for the acceptance of our paper, which is of great significance to us and serves as a strong encouragement. Furthermore, we would be grateful to receive any additional suggestions or feedback that could further strengthen our work and potentially lead to a higher evaluation.
> >
> > Kind regards,
> >
> > Authors

---

### Official Review · Reviewer_PboH · 2025-06-30

**Clarity:** 2
**Significance:** 2
**Originality:** 2
**Rating:** 4
**Confidence:** 3

**Summary:**

This paper focuses on addressing the hallucination issue in LLMs by optimizing causal relations. The approach first identifies causal knowledge and then optimizes it using reinforcement learning. Experiments show that it performs well on several NLU tasks.

**Questions:**

1. I think "enhancing model causality while preserving valuable pretrained knoweldge" is closedly related to continual leanring. I strongly recomment the author to add a related work section on continual leanrning (see above W1 and potential reference)

2. Could you expand the evaluation? The current evaluation is relatively simple and could benefit from broader or more challenging benchmarks (see W2 above).

**Ethical Concerns:**

["NO or VERY MINOR ethics concerns only"]

**Final Justification:**

The related work and additional experiemnts addresed my concerns. I raised score accordingly.

**Limitations:**

No discussion in the main content. Perhaps the author can add some cost analysis

**Quality:**

2

**Strengths And Weaknesses:**

**Strengths:**
1. It is an important problem to solve: avoid the hallucination where the output is based on spurious relations
2. The writing is generally clear and the figure is informative

**Weaknesses:**

1. The "locate and optimize" strategy has been widely used in the continual learning community [1, 2, 3]. It would be helpful if the authors could compare their approach with these existing methods to better situate their contribution.
2. The authors should consider applying their method to LLMs and evaluating it on more complex tasks. The current models and datasets listed in Table 1 are too simple (i.e., SoTA LLM can easily achieve 90%+ acc.), which limits the informativeness of the evaluation.

[1]: Continual Pre-training of Language Models, ICLR 2023.
[2]: Locating and Editing Factual Associations in GPT, NeurIPS 2022.
[3]: Continual learning of natural language processing tasks: A survey， 2022

---

> ### Author Rebuttal · Authors · 2025-07-31
>
> Dear Reviewer PboH,
>
> We deeply appreciate your valuable review and insightful comments. We greatly thank you for your recognition of the significance of the problem we address, the quality of our paper, the richness of the figures, and the strong performance of our approach across various tasks. Below are our detailed responses to each of your concerns:
>
> > **Q1: Relation to continual learning.**
>
> We fully agree with you that enhancing model causality while preserving valuable pretrained knowledge has commonalities with the idea in continual learning, which aims to improve performance on new tasks without forgetting previous ones. Inspired by your valuable feedback, we have conducted extensive research and compared our approach with related methods as follows:
>
> - Several knowledge editing approaches follow a two-step paradigm of localization followed by optimization [1,2], which have been involved in the **“Knowledge Localization” section of Related Work**. These methods typically perturb model parameters or sample activations to identify the location of knowledge by observing changes in model predictions, and then edit only the identified parameters. However, these approaches can **track probability changes for individual tokens only at a time, and are primarily suited to structured triplet data, restricting their applicability to more general or unstructured language scenarios**. Furthermore, as noted in **Lines 64–67**, these methods perform localization at the layer level based on predefined matrix types (e.g., in feed-forward networks), which is often coarse-grained and inflexible. In contrast, our method employs a dynamically learned localization network to perform fine-grained localization at the matrix level, offering greater precision and adaptability.
> - Another related line of work focuses on selecting which weight matrices to update for new tasks based on their importance. However, these methods often rely on a single metric, such as gradients, for soft masking or apply masking to predefined modules, resulting in limited accuracy and flexibility [3,4]. Additionally, continual learning methods primarily aim to learn new tasks without forgetting previous ones, but they **cannot explicitly model or enhance the causal relationships within the model**. In contrast, our approach identifies causal-sensitive parameter matrices by analyzing behavioral differences across original and augmented counterfactual samples using a dynamically learned localization network. These identified parameters are then optimized using an enhanced policy optimization algorithm guided by four reward signals: accuracy, robustness, calibration, and confidence, thereby improving the model's generalization and robustness.
>
> To more clearly delineate the differences between our approach and those in continual learning, we have **added the following content to the Related Work section**: “*Some knowledge editing approaches utilize parameter-based or activation-based techniques for knowledge localization [21,62,64], as outlined before, wherein only the identified parameters are subsequently updated. However, these methods are typically constrained to tracking the probability shift of a single token at a time and are primarily designed for structured triplet-style data, which limits their applicability to more general or unstructured language scenarios. Moreover… Another related line of work focuses on selecting which weight matrices to update for new tasks based on their importance [45,46]. However, most of these methods rely on single metrics like gradients for soft masking or apply masking to predefined modules, which provides a limited perspective. Moreover, these approaches primarily aim to learn new tasks while preserving knowledge in old ones, but they cannot explicitly model or enhance the model’s internal causal relationships. In contrast, our framework…*”
>
> > **Q2: Experiments on more tasks and LLMs.**
>
> The experiments in our manuscript cover three important tasks, including **debiasing, hallucination mitigation, and out-of-distribution generalization**, using **twelve LLMs ranging in size from 18M to 30B parameters**. To ensure fair and meaningful comparisons, the experimental settings for Table 1—including datasets and model selections—follow those used in previous SOTA debiasing studies [5,6].
>
> In response to your valuable suggestion, we have extended our evaluation to more challenging tasks and LLMs.
>
> - First, we conduct additional evaluations on two challenging hallucination datasets, including **FACTOR** and **HaluEval-Sum**, using LLaMA-2-7B and LLaMA-3-8B models.
>   - FACTOR [7] focuses on context consistency and measures the tendency of language models to generate factual information. It is a text completion task where the goal is to identify the correct completion from nonfactual statements given a prefix. The task includes two datasets from different sources: Wiki-FACTOR (2994 examples) and News-Factor (1036 examples). Factuality is assessed by whether the model assigns the highest likelihood to the factually correct completion over the other options.
>   - HaluEval-Sum [8] provides texts paired with both hallucinated and correct responses. We use its summarization track to evaluate the truthfulness of LLMs on longer sequences, consisting of 10,000 samples. For each sample, LLMs are tasked with determining whether the provided summary contains non-factual or hallucinated information relative to the given document. We report accuracy for both hallucinated and correct summaries, including arithmetic mean accuracy (Acc-A) and harmonic mean accuracy (Acc-H).
> |Dataset|FACTOR||HaluEval||
> |:-|:-:|:-:|:-:|:-:|
> ||Wiki|News|Acc-A|Acc-H|
> |LLaMA-2-7B|58.64|72.25|48.01|19.92|
> |+DoLa|60.13|72.86|48.65|27.78|
> |+SH2|64.10|73.59|50.49|50.50|
> |+CDRO|**65.29**|**74.76**|**51.58**|**51.66**|
> |LLaMA-3-8B|65.59|78.42|55.07|28.95|
> |+DoLa|68.90|79.71|54.92|30.43|
> |+SH2|70.03|80.55|60.38|53.81|
> |+CDRO|**71.05**|**82.45**|**61.27**|**55.03**|
> - Subsequently, we provide detailed comparison results for hallucination mitigation between our method and the best-performing baselines (HaDeMiF, ICLR 2025) on five NLG tasks—NQ, SciQ, TriviaQA, TruthfulQA, and WikiQA—evaluated using the **LLaMA-3-8B, LLaMA-2-13B, Qwen-2.5-7B, and LLaMA-30B** models.
> |Task|NQ||SciQ||TriviaQA||TruthfulQA||WikiQA||
> |:-|:-:|:-:|:-:|:-:|:-:|:-:|:-:|:-:|:-:|:-:|
> |Metric|Acc@50(↑)|Cov@50(↑)|Acc@50(↑)|Cov@90(↑)|Acc@50(↑)|Cov@60(↑)|Acc@50 (↑)|Cov@40(↑)|Acc@50 (↑)|Cov@50 (↑)|
> |LLaMA-3-8B|||||||||||
> |HaDeMiF|0.542|0.516|0.892|0.459|0.620|0.369|0.590|0.721|0.620|0.458|
> |CDRO|**0.580**|**0.625**|**0.918**|**0.490**|**0.658**|**0.405**|**0.620**|**0.755**|**0.650**|**0.580**|
> |LLaMA-2-13B|||||||||||
> |HaDeMiF|0.491|0.415|0.842|0.405|0.486|0.270|0.441|0.613|0.655|0.434|
> |CDRO|**0.518**|**0.520**|**0.885**|**0.440**|**0.517**|**0.305**|**0.512**|**0.648**|**0.720**|**0.520**|
> |Qwen-2.5-7B|||||||||||
> |HaDeMiF|0.588|0.550|0.909|0.492|0.688|0.421|0.646|0.760|0.692|0.532|
> |CDRO|**0.613**|**0.650**|**0.928**|**0.530**|**0.713**|**0.478**|**0.668**|**0.805**|**0.724**|**0.630**|
> |LLaMA-30B|||||||||||
> |HaDeMiF|0.512|0.503|0.899|0.435|0.469|0.264|0.542|0.697|0.462|0.218|
> |CDRO|**0.529**|**0.620**|**0.903**|**0.470**|**0.495**|**0.315**|**0.562**|**0.737**|**0.478**|**0.402**|
>
> All of the above experiments will be incorporated into the camera-ready version of our manuscript. As demonstrated, despite variations in models and datasets, **our method consistently delivers robust and effective performance**.
>
> > **Q3: No limitation discussion in the main content. Perhaps the author can add some cost analysis.**
>
> Due to space constraints, the limitations are presented in **Section A.13 of the appendix**. Following your valuable suggestion, a condensed version will be incorporated into the main text as part of the “Conclusion and Discussion” section. Moreover, the complexity analysis is presented in **Section A.9 of the appendix**, which indicates that despite the increased time complexity introduced by the adaptive knowledge localization module, our approach demonstrates **superior parameter efficiency** (Table 15, with an improvement of **up to 58%**), thereby achieving a better tradeoff between preserving pretrained knowledge and enhancing downstream task performance. Moreover, it consistently **achieves the best performance** across various tasks and models. Additionally, the logistic model **converges after a limited number of training iterations** (as shown in Figs. 3(b) and 11), allowing the omission of the logistic update step in subsequent training stages, further improving training efficiency. It is worth noting that while our method incurs some additional computational overhead, it remains a highly efficient fine-tuning approach, particularly when compared to the full fine-tuning for LLMs. A condensed cost analysis will be incorporated into Section 4 of the manuscript.
>
> [1] Locating and editing factual associations in gpt. NeurIPS 2022.
>
> [2] Mass-editing memory in a transformer. ICLR 2022.
>
> [3] Continual pre-training of language models. ICLR 2023.
>
> [4] Continual learning of natural language processing tasks: A survey." arXiv preprint arXiv:2211.12701 (2022).
>
> [5] Towards fair decision: A novel representation method for debiasing pre-trained models. Decision Support Systems 2024.
>
> [6] Causal-debias: Unifying debiasing in pretrained language models and fine-tuning via causal invariant learning. ACL 2023.
>
> [7] Generating benchmarks for factuality evaluation of language models. ACL 2024.
>
> [8] HaluEval: A large-scale hallucination evaluation benchmark for large language models. EMNLP 2023.

---

> > ### Author Response · Authors · 2025-08-06
> > **Looking forward to your reply.**
> >
> > Dear Reviewer PboH,
> >
> > We hope that our revisions and clarifications have adequately addressed your concerns. Your constructive feedback has been instrumental in enhancing the quality of our work. Should any aspects still require further elaboration, or if you have any additional questions, we would be pleased to address them. We deeply value your thoughtful feedback and would be truly grateful if the improvements and clarifications made could be taken into account in the reevaluation of our work.
> >
> > Thank you once again for your time and effort in reviewing our work.
> >
> > Kind regards,
> >
> > Authors

---

> ### Comment · Area_Chair_4jwJ · 2025-08-06
>
> Dear Reviewer,
>
> This is a gentle reminder that the Author-Reviewer Discussion period will end on August 8, 11:59pm AoE.
>
> If you have not yet responded to the author’s rebuttal or comments, please take a moment to participate in the discussion. Simply submitting a “Mandatory Acknowledgement” without engaging in any discussion is not sufficient.
>
> Your active participation is important to ensure a fair and transparent review process.
>
> Thank you again for your valuable service.
>
> Best regards,
> AC

---

### Official Review · Reviewer_QZLY · 2025-07-03

**Clarity:** 2
**Significance:** 3
**Originality:** 3
**Rating:** 4
**Confidence:** 4

**Summary:**

This study introduces a novel Causality-driven Robust Optimization (CDRO) approach to enhance the causal reasoning abilities of Large Language Models (LLMs), reducing their reliance on spurious correlations and mitigating hallucinations. CDRO first identifies causality-sensitive parameters in LLMs by comparing training dynamics across original samples, augmented counterfactual samples, and paraphrased variants, using a lightweight logistic regression model. These identified parameters are then optimized with an enhanced policy optimization algorithm, whose reward function promotes both generalization and robustness. Extensive experiments with twelve different LLMs across various tasks demonstrate CDRO's superiority in reducing spurious associations, mitigating hallucinations, alleviating biases, and improving out-of-distribution generalization. The key contributions include a new method for localizing causal knowledge in LLMs, a collaborative optimization framework, and validation of effectiveness through comprehensive experiments.

**Questions:**

1. Given that the study uses prompted LLMs to score constructed counterfactual and paraphrased samples, could this LLM-based evaluation method inherit the inherent biases or limitations of the evaluating models, leading to inconsistent or inaccurate assessments of sample quality? And might the reliance on such model-in-the-loop scoring—especially when using the same or similar LLMs for both generation and evaluation—create closed-loop biases that fail to align with human judgment or real-world task requirements, thereby undermining the effectiveness of the augmented data?
2. Given that the study claims to selectively update Causality-Sensitive Parameters rather than all model parameters, several questions arise regarding the clarity of this process: How exactly are the "causality-sensitive" characteristics of these parameters operationally defined beyond the mentioned gradient differences and cosine similarities of hidden states? Is there empirical evidence to confirm that the localized parameters indeed encode causal relationships rather than other types of task-relevant information? Additionally, how is the threshold for selecting these parameters (e.g., "matrices with predicted probabilities exceeding the average") determined, and does this threshold remain robust across different model architectures or task domains? Finally, since only specific parameter components are updated, is there a risk of creating imbalances in the model's internal representations, and how does the framework mitigate such potential side effects?
3. Given that CDRO aims to enhance causal reasoning by selectively optimizing parameters identified through comparisons of training dynamics across original, counterfactual, and paraphrased samples, how does the framework account for the possibility that causal relationships in real-world scenarios are often complex, multi-faceted, or context-dependent—far more intricate than the binary or simplified causal signals captured by the generated counterfactuals? Moreover, could the reliance on predefined types of augmented samples (counterfactuals and paraphrases) limit the model’s ability to learn causal structures that cannot be easily approximated by such perturbations, and if so, how might the framework be adapted to capture more nuanced causal dependencies?

**Ethical Concerns:**

["NO or VERY MINOR ethics concerns only"]

**Final Justification:**

We thank the authors for their thorough and constructive rebuttal, as well as for the additional experiments and clarifications provided. The new ablations, cross-model validations, and detailed explanations effectively address the reviewers’ core concerns while highlighting the paper’s strengths. After careful consideration, we maintain our initial “Borderline Accept: Technically solid paper where reasons to accept outweigh reasons to reject” rating. We look forward to seeing the minor revisions incorporated in the camera-ready version.

**Limitations:**

yes

**Quality:**

3

**Strengths And Weaknesses:**

**Strengths**:
- Effectively enhances LLMs' causal reasoning abilities, reducing reliance on spurious correlations and mitigating hallucinations across various tasks .
- Outperforms baselines in debiasing (gender and race) and improving OOD generalization, with superior performance on both NLU and NLG tasks.
- Adopts a parameter-efficient optimization strategy, preserving pretrained knowledge while enhancing downstream task performance .
- The collaborative optimization framework (logistic regression + enhanced REINFORCE++) enables dynamic localization of causal components, ensuring adaptability .
- Comprehensive experiments with 12 LLMs validate its robustness and generalizability .

**Weaknesses**:
- Introduces additional time complexity due to the adaptive knowledge localization module, though the trade-off is justified by performance gains .
- The effectiveness of causal component localization heavily depends on the quality of counterfactual and paraphrased samples, posing challenges in data augmentation. Moreover, there is a lack of ablation experiments on counterfactual and paraphrased samples, making the significance of paraphrased samples questionable.
- The layout of tables and figures is suboptimal. In the appendix, the places where figures and tables are cited are far from their actual positions, which may cause confusion for readers.
- The description of parameter updates for the selected model components is not clear enough, with only limited information provided in the flowcharts.

---

> ### Author Rebuttal · Authors · 2025-07-31
>
> Dear Reviewer QZLY,
>
> We sincerely appreciate your thorough review and insightful comments. Thank you for your recognition of the significance and novelty of our work, the adaptability and parameter efficiency of our framework, the comprehensiveness of our experiments, and the effectiveness of our approach. Below, we provide detailed responses to your concerns:
>
> > **Q1: Additional time complexity.**
>
> As you mentioned, despite the added complexity, our approach achieves superior parameter efficiency—improving by **up to 58%** (Table 15)—striking a better balance between preserving pretrained knowledge and enhancing downstream task performance. Moreover, it consistently **attains top performance** across diverse tasks and models. Furthermore, the logistic model **converges after a limited number of iterations** (Figs. 3(b) and 11), enabling the omission of logistic updates in later training stages, further improving efficiency.
>
> > **Q2: Questions about data collection.**
>
> We humbly respond to your concerns from the following perspectives:
> - To minimize costs and leverage the established capability of LLMs [1,2] to generate counterfactual and paraphrased samples, the entire data collection process is conducted by prompting off-the-shelf LLMs. Rigorous design and careful considerations ensure the reliability of the generated data.
>   - We utilize **advanced LLMs** for the data collection process, including GPT-4o and LLAMA-3-70B, which exhibit high stability and reliability in generation tasks [3,4].
>   - We **generate eight counterfactual and paraphrased variants per sample and evaluate each three times across multiple dimensions**. The **highest-scoring** variant is then selected for downstream use, mitigating randomness and errors inherent in single generations.
>   - Despite potentially introducing biases, LLM-based evaluation remains a widely adopted practice. To improve reliability, **we employ carefully designed prompts and conduct multiple independent evaluations per sample**. Additionally, we compare evaluations using the same LLM as for generation against a different LLM. From Table 6, the two settings yield comparable and satisfactory results, with a Pearson correlation of 0.84, indicating **reliable assessment of the generated samples**.
>   - As noted in Footnote 5, the application of the generated samples is weighted by their evaluation scores, ensuring that **higher-quality samples have a more dominant influence**.
> - CDRO consistently improves performance across diverse tasks and models, as indicated in our experiments. Since **both localization and fine-tuning rely on the augmented data, these gains indirectly validate the quality of the generated samples**.
> - Motivated by your feedback, we conducted ablation studies to assess the impact of counterfactual and paraphrased samples. Results show that **omitting either leads to reduced performance**, confirming their effectiveness. Besides, from Figs. 3(b) and 11, all indicator coefficients are non-zero at model convergence, demonstrating each indicator’s contribution and highlighting the importance of jointly considering counterfactual and paraphrased samples.
> |Task|NQ||SciQ||TriviaQA||TruthfulQA||WikiQA||
> |:-|:-:|:-:|:-:|:-:|:-:|:-:|:-:|:-:|:-:|:-:|
> |Metric|Acc@50(↑)|Cov@50(↑)|Acc@50(↑)|Cov@90(↑)|Acc@50(↑)|Cov@60(↑)|Acc@50 (↑)|Cov@40(↑)|Acc@50 (↑)|Cov@50 (↑)|
> |CDRO|**0.376**|**0.228**|**0.781**|**0.245**|**0.520**|**0.258**|**0.456**|**0.529**|**0.665**|**0.627**|
> |W/o paraphrase|0.375|0.224|0.777|0.243|0.519|0.255|0.450|0.528|0.662|0.624|
> |W/o counterfactual|0.359|0.220|0.769|0.233|0.505|0.247|0.436|0.514|0.657|0.621|
>
> > **Q3: Layout of tables and figures in the appendix.**
>
> We apologize for the improper placement of figures and tables. The appendix has been revised to **position them as close as possible to their corresponding citations**.
>
> > **Q4: Parameter updates for the selected components.**
>
> We apologize for the previous lack of clarity. **Section 3.3 (Lines 196-238)** details the updates of the selected components, which are performed using an enhanced REINFORCE++ algorithm. The optimization objective in Eq. (2) incorporates a composite reward function combining accuracy, robustness, calibration, and confidence. Moreover, only the low-rank principal components of these matrices are updated following PiSSA [5] to improve efficiency.
>
> To enhance clarity, Section 3.3 has been **reorganized into two subsections**: "*Optimization Process for the Selected Components of LLMs*" and "*Optimization Process for the Logistic Model*". Each subsection **begins with a topic sentence summarizing the corresponding optimization strategy**. We have also added more details about the optimization of LLMs in Fig.2.
>
> > **Q5: Selection of causality-sensitive parameters.**
>
> We respectfully address your concerns from the following perspectives:
> - Our motivation for identifying causal-sensitive parameters is based on the idea that **such parameters exhibit behavioral differences, such as loss gradients and activation values, between original and counterfactual samples**. This approach is consistent with prior localization methods that analyze prediction changes induced by perturbations to parameters or activations [6,7]. However, manually determining the influence of various indicators on localization is unreasonable. To address this, we propose utilizing a simple logistic model to automatically learn this relationship. Future work may extend this by incorporating more sophisticated indicators, such as projections of activations onto model parameters or the Fisher information matrix, to enhance identification accuracy.
> - As shown in **Figs. 3(b) and 11**, the relationship between causal localization and selected indicators aligns with our expectations. Specifically, gradient difference coefficients are positive, whereas hidden state cosine similarity coefficients are negative. These results confirm that **the identified parameters exhibit distinct behaviors between original and counterfactual samples, supporting their sensitivity to causal relationships**.
> - Naturally, relying solely on original and paraphrased samples may cause localized parameters to capture superficial statistical patterns. **Ablation results in Q2** show that excluding counterfactuals leads to a significantly larger performance decline than excluding paraphrased samples, indicating that counterfactual augmentations enable **localized parameters to capture causal relationships rather than statistical correlations**.
> |Dataset|SST-2||MNLI||QQP||
> |:-|:-:|:-:|:-:|:-:|:-:|:-:|
> |OOD data|IMDB-Cont|IMDB-CAD|HANS|AdvNLI|PAWS|
> |Average|**89.62**|**92.65**|**77.68**|**39.40**|**46.01**|
> |Top-15|89.19|92.37|77.54|38.98|45.43|
> |0.5|89.23|92.48|77.35|38.92|45.66|
> - The threshold for selecting causal components is set as the average logistic probability across all parameter matrices, with matrices having probabilities above this threshold being updated. Due to the trainable nature of the logistic model, this threshold is dynamic. We compare three threshold strategies—Top-15 matrices, matrices with probabilities $> 0.5$, and the average threshold—across two tasks: OOD generalization (RoBERTa-base) and hallucination mitigation (LLaMA-2-7B). **Our dynamic threshold consistently outperforms fixed manual ones, demonstrating robustness across tasks and models**.
> |Task|NQ||SciQ||
> |:-|:-:|:-:|:-:|:-:|
> |Metric|Acc@50(↑)|Cov@50(↑)|Acc@50(↑)|Cov@90(↑)|
> |Average|**0.376**|**0.228**|**0.781**|**0.245**|
> |Top-15|0.373|0.225|0.775|0.238|
> |0.5|0.369|0.226|0.778|0.240|
> - CDRO **avoids internal model imbalance by dynamically updating different matrices throughout training rather than fixing them**. As shown in **Figs. 3(a) and 13**, updates occur across various layers and matrix types, albeit to varying degrees. Moreover, **Figs. 4 and 12** indicate that our CDRO optimization achieves greater stability and higher accuracy compared to standard REINFORCE++.
>
> > **Q6: Complex causal relationships.**
>
> Insightful question! We humbly respond to your concerns from the following aspects:
> - We fully acknowledge the inherent complexity of causal relationships, which often arise from multiple factors—for instance, heart disease may stem from diet, genetics, and more. Nevertheless, **counterfactuals offer a clear and effective approach to model and interpret causal relationships**. While generating counterfactual samples for all possible causal relationships is impractical, integrating causal knowledge from the augmented counterfactuals represents a significant advancement in model learning. Accordingly, we identify causal-sensitive components by comparing parameter behaviors between original and counterfactual samples and update the identified ones. **As shown in Figs. 4 and 12, CDRO enhances both stability and effectiveness**.
> - We utilize an enhanced policy optimization method to update the identified components, with the reward incorporating multiple metrics. The observed gains in various tasks demonstrate that incorporating counterfactual samples **strengthens both the model’s causal reasoning and robustness**.
> - Your feedback has **inspired further extensions to our framework**. One direction is to generate multiple counterfactuals per original sample, each targeting different aspects, to enable more precise localization and optimization, albeit with increased augmentation complexity. Additionally, conducting counterfactual augmentation in the deep semantic space could help capture more nuanced causal relationships.
>
> [1] Autocad: Automatically generate counterfactuals for mitigating shortcut learning.
>
> [2] Prompting large language models for counterfactual generation: An empirical study.
>
> [3] The llama 3 herd of models.
>
> [4] Gpt-4o system card.
>
> [5] Pissa: Principal singular values and singular vectors adaptation of large language models.
>
> [6] Locating and editing factual associations in gpt.
>
> [7] Mass-editing memory in a transformer.

---

> > ### Author Response · Authors · 2025-08-06
> > **Looking forward to your reply.**
> >
> > Dear Reviewer QZLY,
> >
> > We hope that our explanations and revisions have adequately addressed your concerns. We sincerely appreciate your thoughtful feedback, which has been invaluable in enhancing the quality of our paper. Should any points remain unclear, or if you have any further suggestions, please do not hesitate to share them. We deeply value your insightful feedback and would be deeply appreciative if you could take our clarifications and improvements into account when reassessing our work.
> >
> > Thank you once again for your time and effort in reviewing our work.
> >
> > Kind regards,
> >
> > Authors

---

> > ### Comment · Area_Chair_4jwJ · 2025-08-06
> >
> > Dear Reviewer,
> >
> > This is a gentle reminder that the Author-Reviewer Discussion period will end on August 8, 11:59pm AoE.
> >
> > If you have not yet responded to the author’s rebuttal or comments, please take a moment to participate in the discussion. Simply submitting a “Mandatory Acknowledgement” without engaging in any discussion is not sufficient.
> >
> > Your active participation is important to ensure a fair and transparent review process.
> >
> > Thank you again for your valuable service.
> >
> > Best regards,
> > AC

---

### Official Review · Reviewer_z2Ph · 2025-07-03

**Clarity:** 2
**Significance:** 2
**Originality:** 3
**Rating:** 3
**Confidence:** 3

**Summary:**

This paper proposes a new robust optimization framework based on causality, called CDRO, aiming to reduce LLMS 'reliance on false correlations and enhance their resilience across various tasks.   Specifically, the method proposed in this work first identifies the parameter components that capture causal relationships by analyzing the training dynamics in the weight matrices of the original samples, counterfactual samples, and paraphrase samples.   These dynamics are modeled through the logistic regression mechanism, making it possible to automatically and adaptively locate the parameters related to causal relationships.   Meanwhile, to further improve and optimize the process, the author introduced a collaborative reinforcement learning strategy, alternately updating the identified causal parameters and the logistic regression model.   A large number of experiments on various NLU and NLG tasks have shown that CDRO consistently outperforms the comparison baseline in reducing false correlations, suppressing knowledge illusions, and improving overall model performance.

**Questions:**

My question is recorded in the weakness section

**Ethical Concerns:**

["NO or VERY MINOR ethics concerns only"]

**Final Justification:**

Overall, I think this paper is between borderline reject and borderline accept. The biggest concern is that it relies on a more advanced instruction-following LLM.

**Limitations:**

Refer to the weakness.

**Quality:**

2

**Strengths And Weaknesses:**

Strengths The motivation of this paper is clear and the writing is concise and of high quality.   The experimental section is relatively comprehensive, with extensive evaluations conducted on the proposed method.   Additionally, the method has been compared with various baseline approaches, further highlighting its effectiveness.

Weaknesses 1.  The author mentioned that LLMS have counterfactual issues.   However, during the data construction stage, prompt LLMS are used for data generation.   Does this mean that the data quality depends on the LLM itself?   In other words, how can we ensure that the generated data is trustworthy?

2.  The author mentioned that other methods have fine-tuned all the parameters of the model, but the method proposed by the author only fine-tunes some parameters.   Efficient parameter fine-tuning methods are common.   What are the general implementation methods of other baselines like?   Is there really no method that only fine-tunes some parameters?

3.  In the experimental section, why weren't all the baselines compared on some datasets or some models?

4.   Has the author conducted experiments on datasets for hallucinations?   Hallucination evaluation on the NLG task may not well reflect the hallucination problem of the model.

---

> ### Author Rebuttal · Authors · 2025-07-31
>
> Dear Reviewer z2Ph,
>
> We sincerely thank you for your thoughtful review and constructive comments. We greatly appreciate your recognition of the clarity of our research motivation, the quality of our writing, the comprehensiveness of our experiments, and the superiority of our performance. Below are our detailed responses to each of your concerns:
>
> > **Q1: The trustworthiness of the generated data.**
>
> To minimize costs and leverage the established capability of LLMs [1,2] to generate counterfactual and paraphrased samples, the entire data collection process is conducted by directly prompting off-the-shelf LLMs. Rigorous design and careful considerations ensure the reliability of the generated data.
> - We utilize **advanced instruction-following LLMs** for the data collection process, including GPT-4o and LLaMA-3-70B, which exhibit high stability and reliability in generation tasks [3,4].
> - To minimize potential biases, we **generate multiple counterfactual and paraphrased samples** for each original sample. These are then **evaluated three times across multiple dimensions**, including alignment or divergence between the answers of augmented and original samples, answer correctness, thematic consistency, clarity, and safety and privacy. Specifically, we generate eight independent samples per original input and **select the highest-scoring one** for downstream use, which mitigates the randomness or errors inherent in single-generation results.
> - We evaluate the performance of using the same LLM for both generation and evaluation, as well as employing a different LLM for evaluation. As shown in Table 6, both approaches yield comparable and satisfactory performance. Moreover, the Pearson correlation coefficient between the scores produced by the two approaches is 0.84. These findings suggest that **either method can effectively evaluate the generated samples**.
> - As noted in Footnote 5, the application of the generated samples is weighted by their evaluation scores, thereby ensuring that **higher-quality samples have a more dominant influence**.
> - CDRO consistently improves performance across diverse tasks and models, as demonstrated in our experiments. Since **both localization and fine-tuning rely on the augmented data, these gains indirectly validate the quality of the generated samples**.
> - We further conduct ablation experiments to demonstrate the effectiveness of both generated counterfactual and paraphrased samples. The results demonstrate that **omitting either augmentation leads to performance degradation**, highlighting the essential contribution of each.
> |Task|NQ||SciQ||TriviaQA||TruthfulQA||WikiQA||
> |:-|:-:|:-:|:-:|:-:|:-:|:-:|:-:|:-:|:-:|:-:|
> |Metric|Acc@50(↑)|Cov@50(↑)|Acc@50(↑)|Cov@90(↑)|Acc@50(↑)|Cov@60(↑)|Acc@50 (↑)|Cov@40(↑)|Acc@50 (↑)|Cov@50 (↑)|
> |CDRO|**0.376**|**0.228**|**0.781**|**0.245**|**0.520**|**0.258**|**0.456**|**0.529**|**0.665**|**0.627**|
> |W/o paraphrase|0.375|0.224|0.777|0.243|0.519|0.255|0.450|0.528|0.662|0.624|
> |W/o counterfactual|0.359|0.220|0.769|0.233|0.505|0.247|0.436|0.514|0.657|0.621|
>
> > **Q2: Other methods that only fine-tune partial parameters.**
>
> We humbly address your concerns from the following perspectives:
> - Existing debiasing and hallucination mitigation methods (e.g., Causal-Debiasing [5], PCFR [6], HaDeMiF [7]) typically rely on **indiscriminate fine-tuning of all model parameters, without distinguishing between components that encode genuine causal knowledge and those that are most important for debiasing**.
> - As you mentioned, some parameter-efficient fine-tuning (PEFT) methods also update only a subset of model parameters, typically falling into two categories. The first is **selective PEFT**, which targets parameters deemed important for downstream tasks. However, such methods typically predefine fixed parameter types (e.g., bias-only) [8] or preselect a parameter subset that remains unchanged throughout training [9], thereby limiting adaptability. The second category comprises **low-rank adaptation methods** (e.g., LoRA [8], PiSSA [9]), which constrain updates to low-rank matrices but apply them uniformly across all weight matrices, regardless of task-specific relevance. In contrast, our proposed CDRO framework **combines parameter efficiency with causality-aware selectivity**. It dynamically identifies and fine-tunes only those parameter matrices most critical for encoding causal relationships, thereby enhancing causal reasoning while minimizing redundant updates.
> - The 'Label Smoothing' baseline in Table 2 refers to fine-tuning with label smoothing loss in combination with LoRA, which yields suboptimal performance. To further highlight **the superior effectiveness of our method**, we conduct empirical comparisons with vanilla LoRA, PiSSA, and BitFit, all of which demonstrate poor performance on hallucination mitigation tasks.
> |Task|NQ||SciQ||TriviaQA||TruthfulQA||WikiQA||
> |:-|:-:|:-:|:-:|:-:|:-:|:-:|:-:|:-:|:-:|:-:|
> |Metric|Acc@50(↑)|Cov@50(↑)|Acc@50(↑)|Cov@90(↑)|Acc@50(↑)|Cov@60(↑)|Acc@50 (↑)|Cov@40(↑)|Acc@50 (↑)|Cov@50 (↑)|
> |LoRA|0.201|0.060|0.215|0.006|0.289|0.015|0.176|0.000|0.257|0.000|
> |PiSSA|0.203|0.061|0.209|0.001|0.302|0.018|0.180|0.002|0.278|0.003|
> |BitFit|0.197|0.054|0.210|0.002|0.267|0.010|0.174|0.000|0.260|0.000|
> |CDRO|**0.376**|**0.228**|**0.781**|**0.245**|**0.520**|**0.258**|**0.456**|**0.529**|**0.665**|**0.627**|
> - To further clarify the distinction between PEFT methods that fine-tune partial parameters and our proposed approach, we have added the following content to the **Related Work** section: “*Some parameter-efficient fine-tuning (PEFT) methods also update only a subset of model parameters. For example, selective PEFT methods [45,53] aim to fine-tune parameters most relevant to downstream tasks. However, they often predefine fixed parameter types (e.g., bias terms) or preselect a parameter subset based on a single static criterion, which remains unchanged throughout training and thereby limits adaptability. Moreover, low-rank adaptation methods [61,65] reduce trainable parameters by restricting updates to low-rank matrices, but apply such modifications uniformly across all weights, without accounting for task-specific importance. In contrast, our method…*”
>
> > **Q3: Compared baselines.**
>
> To the best of our knowledge, our study is **the first to integrate debiasing, hallucination mitigation, and out-of-distribution (OOD) generalization into a unified experimental framework**. In contrast, prior studies typically address these tasks independently, with baselines exhibiting limited generalizability and challenges in cross-task deployment. To comprehensively evaluate our approach, we compare it against SOTA methods specialized for each task. Notably, during experimental investigation, we have tried to extend certain methods, such as Causal-Debias [5] and PCFR [6], which are originally designed for debiasing, to enhance OOD generalization (Tables 3 and 10). However, since both methods can only address specific types of bias (i.e., gender and race) and require full model fine-tuning, their applicability to LLMs for hallucination mitigation is limited.
>
> > **Q4: More experiments for hallucination mitigation.**
>
> Following prior research [7], we evaluate the performance of our approach for hallucination mitigation using five representative NLG benchmarks (NQ, SciQ, TriviaQA, TruthfulQA, and WikiQA) across eight LLMs ranging from 1.5B to 30B parameters. The results are presented in Table 2 and Fig. 10 of our manuscript.
>
> However, we fully acknowledge that additional experiments could further substantiate the effectiveness of our approach. Consequently, we conduct **further evaluations using LLaMA-2-7B and LLaMA-3-8B models on two hallucination datasets: FACTOR and HaluEval-Sum**.
> - FACTOR [12] focuses on context consistency and measures the tendency of LLMs to generate factual information. The task includes two datasets: Wiki-FACTOR (2994 examples) and News-Factor (1036 examples). Factuality is assessed by whether the model assigns the highest likelihood to the factually correct completion over the other options.
> - HaluEval-Sum [13] provides texts paired with both hallucinated and correct responses. For each sample, LLMs are tasked with determining whether the provided summary contains hallucinated information relative to the given document. We report accuracy for both hallucinated and correct summaries, including arithmetic mean accuracy (Acc-A) and harmonic mean accuracy (Acc-H).
>
> |Data|FACTOR||HaluEval||
> |:-|:-:|:-:|:-:|:-:|
> ||Wiki|News|Acc-A|Acc-H|
> |LLaMA-2-7B|58.64|72.25|48.01|19.92|
> |+DoLa|60.13|72.86|48.65|27.78|
> |+SH2|64.10|73.59|50.49|50.50|
> |+CDRO|**65.29**|**74.76**|**51.58**|**51.66**|
> |LLaMA-3-8B|65.59|78.42|55.07|28.95|
> |+DoLa|68.90|79.71|54.92|30.43|
> |+SH2|70.03|80.55|60.38|53.81|
> |+CDRO|**71.05**|**82.45**|**61.27**|**55.03**|
>
> The results unequivocally demonstrate the superior performance of the proposed CDRO approach in hallucination elimination.
>
> [1] Autocad: Automatically generate counterfactuals for mitigating shortcut learning.
>
> [2] Prompting large language models for counterfactual generation: An empirical study.
>
> [3] The llama 3 herd of models.
>
> [4] Gpt-4o system card.
>
> [5] Causal-debias: Unifying debiasing in pretrained language models and fine-tuning via causal invariant learning.
>
> [6] Towards fair decision: A novel representation method for debiasing pre-trained models.
>
> [7] Hademif: Hallucination detection and mitigation in large language models.
>
> [8] BitFit: Simple parameter-efficient fine-tuning for transformer-based masked language models.
>
> [9] Training neural networks with fixed sparse masks.
>
> [10] Lora: Low-rank adaptation of large language models.
>
> [11] Pissa: Principal singular values and singular vectors adaptation of large language models.
>
> [12] Generating benchmarks for factuality evaluation of language models.
>
> [13] HaluEval: A large-scale hallucination evaluation benchmark for large language models.

---

### Comment · Area_Chair_4jwJ · 2025-08-05
**Reminder: Author-Reviewer Discussion Ends Aug 6**

Dear Reviewers,

This is a quick reminder that the Author-Reviewer Discussion phase (July 31 – Aug 6) is ending soon. Feel free to raise any comments or questions, these can help prompt author responses.

Best,
Area Chair

---

### Note · Authors · 2025-08-12

Dear area chairs and reviewers,

We sincerely appreciate the significant time and effort you have devoted to the review process, as well as your insightful and valuable feedback, which has been instrumental in refining and enhancing our manuscript. We are also truly grateful for the reviewers’ positive recognition of our work. Specifically, the reviewers found that **our research motivation is clear and significant** (Reviewers z2Ph, QZLY, PboH, and 6STb), **the proposed method is novel, well-founded, and adaptable** (Reviewers z2Ph, QZLY, and 6STb), **the experimental evaluation is comprehensive and diverse** (Reviewers z2Ph, QZLY, and 6STb), **the performance is strong and promising** (Reviewers z2Ph, QZLY, PboH, and 6STb), and **the writing is clear and of high quality** (Reviewers z2Ph, PboH, and 6STb).

We have carefully considered and fully addressed each of the reviewers' concerns by providing further clarifications and conducting additional experiments in our response. We sincerely appreciate the acknowledgment from Reviewers z2Ph and 6STb that our responses have effectively addressed their concerns. In light of this, **we would be profoundly grateful if all reviewers would carefully consider the revisions and clarifications we have made when re-evaluating our manuscript and assigning the final scores**.

Once again, we wish to convey our sincere gratitude to all of you for your invaluable contributions to our work.



Kind regards,

Authors

---

### Decision · Program_Chairs · 2025-09-17

**Decision:**

Accept (poster)

**Comment:**

This paper proposes Causality-driven Robust Optimization (CdRO), a principled method that dynamically identifies and selectively updates causal-sensitive components in large language models, thereby reducing spurious correlations and hallucinations while preserving pretrained knowledge and achieving strong robustness across diverse tasks.

**Comments:**

- Reviewer z2Ph (Final Rating: 3 - Borderline reject): Finds the paper clearly written with comprehensive experiments and strong baselines, though remains cautious about data reliability, baseline coverage, and hallucination evaluation; **overall takes a neutral stance between borderline reject and accept.**

- Reviewer QZLY (Final Rating: 4 - Borderline accept): Praises the method’s effectiveness in improving causal reasoning and reducing hallucinations with robust validation, while noting added complexity and missing ablations on paraphrased samples.

- Reviewer PboH (Final Rating: 4 - Borderline accept): Acknowledges the importance of the problem and clarity of presentation, but points out overlap with existing “locate and optimize” strategies and the need for more challenging evaluation datasets.

- Reviewer 6STb (Final Rating: 4 - Borderline accept): Highlights the motivation, parameter efficiency, and clear presentation as strengths, while raising questions on scalability, stability, and potential failure cases.

**Meta Comments:**

Overall, the reviewers agree that this paper addresses an important and timely challenge, reducing spurious correlations and hallucinations in large language models, through a well-motivated and parameter-efficient framework. The proposed Causality-driven Robust Optimization (CdRO) method is clearly presented, technically solid, and validated with extensive experiments across multiple LLMs. While some concerns remain regarding data reliability, broader baseline comparisons, and ablations, the paper’s contributions are viewed as meaningful, its methodology sound, and its motivation compelling.

Taken together, the consensus is positive, and the paper is considered suitable for acceptance.